# Gal3 Plays a Deleterious Role in a Mouse Model of Endotoxemia

**DOI:** 10.3390/ijms23031170

**Published:** 2022-01-21

**Authors:** Juan Carlos Fernández-Martín, Ana María Espinosa-Oliva, Irene García-Domínguez, Isaac Rosado-Sánchez, Yolanda M. Pacheco, Rosario Moyano, José G. Monterde, José Luis Venero, Rocío M. de Pablos

**Affiliations:** 1Departamento de Bioquímicay Biología Molecular, Facultad de Farmacia, Universidad de Sevilla, 41012 Sevilla, Spain; jcfermar3@gmail.com (J.C.F.-M.); irenegarcia391@gmail.com (I.G.-D.); jlvenero@us.es (J.L.V.); depablos@us.es (R.M.d.P.); 2Instituto of Biomedicina de Sevilla (IBiS), Hospital Universitario Virgen del Rocío (HUVR)/CSIC/Universidad de Sevilla, 41012 Sevilla, Spain; Isaac.RosadoSanchez@bcchr.ca (I.R.-S.); ypacheco-ibis@us.es (Y.M.P.); 3Departamento de Farmacología, Toxicología, Medicina Legal y Forense, Universidad de Córdoba, 14014 Córdoba, Spain; r.moyano@uco.es; 4Departamento de Anatomía y Anatomía Patológica Comparadas, Universidad de Córdoba, 14014 Córdoba, Spain; jg.monterde@uco.es

**Keywords:** endotoxemia, galectin-3, lipopolysaccharide, electron microscopy, sepsis

## Abstract

Lipopolysaccharide (LPS)-induced endotoxemia induces an acute systemic inflammatory response that mimics some important features of sepsis, the disease with the highest mortality rate worldwide. In this work, we have analyzed a murine model of endotoxemia based on a single intraperitoneal injection of 5 mg/kg of LPS. We took advantage of galectin-3 (Gal3) knockout mice and found that the absence of Gal3 decreased the mortality rate oflethal endotoxemia in the first 80 h after the administration of LPS, along with a reduction in the tissular damage in several organs measured by electron microscopy. Using flow cytometry, we demonstrated that, in control conditions, peripheral immune cells, especially monocytes, exhibited high levels of Gal3, which were early depleted in response to LPS injection, thus suggesting Gal3 release under endotoxemia conditions. However, serum levels of Gal3 early decreased in response to LPS challenge (1 h), an indication that Gal3 may be extravasated to peripheral organs. Indeed, analysis of Gal3 in peripheral organs revealed a robust up-regulation of Gal3 36 h after LPS injection. Taken together, these results demonstrate the important role that Gal3 could play in the development of systemic inflammation, a well-established feature of sepsis, thus opening new and promising therapeutic options for these harmful conditions.

## 1. Introduction

Systemic inflammation is a major pathogenic feature of different acute and chronic diseases, including sepsis [1]. Indeed, lipopolysaccharide (LPS)-induced endotoxemia leads to overall activation of monocytes and peripheral macrophages to cause peripheral inflammation. At appropriate LPS doses, animals exhibit physiological and biochemical features that mimic certain fulminant forms of Gram-negative bacterial infection along with important hallmarks of sepsis [2]. In fact, LPS-induced endotoxemia induces arterial hypotension, lactic acidosis, tachycardia, and specific temporal elevations of circulating levels of tumor necrosis factor (TNF)-α, interleukin (IL)-6, and high mobility group box (HMGB)-1 [2], which are important features of sepsis and septic shock. In severe conditions, sepsis/endotoxemia can lead to a systemic inflammatory response syndrome (SIRS) that induces a cytokine storm-induced syndrome, a life-threating condition [3]. The cytokine storm is triggered by leukocyte activation after binding of pathogen-associated molecular patterns (PAMPs) such as bacterial, fungal, or viral components or damage-associated molecular patterns (DAMPs) such as endogenous nucleic acid material and proteins [4]. The recognition of PAMPs and DAMPs is mediated through binding to pattern recognition receptors (PRRs), from which the Toll-like receptors (TLRs) play a critical role [4]. In fact, most studied PRRs are TLRs, especially TLR4, the receptor for LPS [5], a wall compound of Gram-negative bacteria widely used to model the acute inflammatory response associated with early sepsis [6].

Endogenous molecules that regulate PRR signaling, and particularly TLR4 signaling, are quite attractive means to preventing or attenuating the cytokine storm associated withsepsis/endotoxemia conditions. In this sense, galectin-3 (Gal3) is emerging as an interesting pharmacological target to treat diseases related to the immune system, due to the immunoregulatory role that is attributed to this protein and its ability to bind to TLR4 [7]. Gal3 has many physiological functions, including adhesion, apoptosis, cell cycle regulation, and cell-to-cell interactions within the extracellular matrix [8,9,10,11]. Moreover, Gal3 has been associated with several diseases, such as cancer, liver diseases, cardiovascular diseases, periodontal diseases, and fibrotic diseases and has even been proposed as a biomarker in several conditions [12,13,14,15,16]. Indeed, we have recently suggested that Gal3 is an attractive molecule to consider in the regulation of the macrophage-related hyperinflammation phase associated withsevere COVID-19 patients [17].

Galectins are soluble proteins defined by the presence of a carbohydrate-recognition domain (CRD), which binds N-acetyl-lactosamine-enriched glycoconjugates present on the cell surface or extracellular matrix [18]. This family of proteins consists of at least 15 members in vertebrates, ofwhich Gal3 is the only chimera-type member [18], which is defined by a C-terminal CRD and an N-terminal tail that allows oligomerization [18]. According to cell type and cellular location, Gal3 has different functions. Gal3 can be found intra-(cytoplasm and nucleus) and extracellularly [19]. We have long demonstrated a pro-inflammatory role of Gal3 in microglia (myeloid cells), as evidenced by the fact that absence or inhibition of Gal3 strongly hinders the microglia response to LPS [7], fibrillary β-amyloid [20], and α-synuclein aggregates [21]. Furthermore, it has been shown to be neuroprotective in several in vivo models of neurodegeneration, including global ischemia, intranigral LPS injection, traumatic brain injury, and spinal cord injury [7,22,23]. We and others have demonstrated the ability of Gal3 to govern immune-related functions through binding to different receptors, including the triggering receptor expressed on myeloid cells 2 (TREM2), the insulin-like growth factor receptor 1 (IGFR1), and, importantly, TLR4 [7,20,24]. All these features make Gal3 an attractive target to modulate peripheral immune responses including those associated withsepsis and endotoxemia. Supporting this, circulating Gal3 has been shown to increase in patients suffering from sepsis [25,26]. A detrimental role of Gal3 has been shown in mice undergoing sepsis induced by either cecal ligation and puncture (CLP) [27] or infection with Francisellanovicida [28]. On the other hand, Gal3 has been reported to act as a negative regulator of LPS-induced endotoxemia and inflammation, which was explained by the ability of Gal3 to bind to LPS to further inhibit the endotoxin-associated pro-inflammatory response [29]. However, this view has been challenged by different studies, including the following:(i) Gal3 has been shown to potentiate LPS-induced IL-1 production and enhance monocyte chemotaxis [30]; (ii) the functional relevance of interactions between Gal3 and LPS remains to be clarified [31,32]; (iii) we have identified Gal3 as a master amplifier of the LPS-induced pro-inflammatory response in microglia [7]. Consequently, we have analyzed the effect of Gal3 in the LPS-induced model of endotoxemia (5 mg/kg) including (i) mortality rate, (ii) acute effect of LPS challenge (1 h) on membrane-bound Gal3 on circulating immune cells, (iii) delayed effect of LPS challenge (36 h) on Gal3 expression on peripheral organs, (iv) overall pro-inflammatory and anti-inflammatory status, and (v) peripheral organ integrity. Our study suggests that Gal3 is a powerful modulator of the peripheral immune system, thus emerging as a potential endogenous alarmin regulating early endotoxemia development.

## 2. Results

### 2.1. Genetic Deletion of Gal3 Decreases the Mortality Rate

As a first step, we performed an LPS dose-response study (0, 7.5, 10 and 15 mg/kg) to monitor the survival of the animals for five days (Appendix A). From this analysis, we found that more than 55.5% of the animals died by day 2 following LPS challenge at a dose of 10 and 15 mg/kg (Appendix A). Similar results were found when we used the 7.5 mg/kg dosage, although we found a clear delay in mortality (Appendix A). Based on this study, a dose of 5 mg/kg was selected to evaluate the effect of the genetic ablation of *Gal3* on the mortality induced in our model of LPS-induced endotoxemia. Under these conditions, 90% of WT succumbed in the first 80 h (Figure 1). LPS injection in Gal3KO animals also induced the death of animals, but we observed a significant delay in the death rate and a clear increase in the survival rate (Figure 1; *p* = 0.033), with about 50% of the animals surviving.

### 2.2. LPS Induces Changes in the Gal3 Levels in Immune Cells of Peripheral Blood, Serum and Tissues

To address whether immune cells act as a potential source of Gal3 under conditions of endotoxemia, we determined the early expression (1 h) of such protein on different subsets of immune cells in peripheral blood by FACS after LPS challenge.In control conditions, all the studied cell subsets expressed Gal3 but at very different levels. While a high frequency of innate immune cells, especially monocytes, expressing Gal3 was found, the frequency of adaptive immune cells expressing Gal3 was significantly lower, withthe lowest frequency being that of CD8^+^ T-cells (Figure 2). Under conditions of endotoxemia, the levels of membrane-bound Gal3 statistically decreased in all the immune cell subsets with the exception of CD8^+^ T-cells, CD4^+^ T-cells, and DCs, which only showed a trend (*p* = 0.2, *p* = 0.056 and *p* = 0.056, respectively) to a reduction of levels of Gal3 after the LPS exposure (Figure 2). Under conditions of endotoxemia, the levels of Gal3 bound to the membrane decreased statistically in B and T cells, neutrophils, monocytes, and macrophages. However, CD8^+^ T-cells, CD4^+^ T-cells, and DCs only showed a trend toward a reduction of levels of Gal3 after the LPS exposure (*p* = 0.2, *p* = 0.056 and *p* = 0.056, respectively; Figure 2).

Next, we determined the amount of Gal3 in the serum of WT animals treated with LPS (1 h) and found a decrease of around 80% with respect to the control group (Figure 3A, *p* = 0.00001). Since serum levels of Gal3 were determined very early in response to LPS challenge, we also determined the mRNA levels of *Gal3* in liver, spleen, and peritoneal macrophages 1 h after LPS challenge. As expected, Gal3KO animals did not express *Gal3*, whereas WT animals showed a basal expression of this galectin (Figure 3B–D). No statistical differences were found between WT animals treated with LPS or saline (Figure 3B–D). Since LPS-induced *Gal3* transcription and translation may require longer than 1 h, and considering that circulating Gal3 may spread and be takenup by peripheral organs to elicit tissue-specific immune responses including Gal3 upregulation, we next analyzed Gal3 protein expression by immunohistochemistry in liver, lung, and spleen 36 h after LPS challenge. Notably, under these conditions, a robust increase of Gal3-expressing cells was found in the examined organs from WT animals ranging from 2.1 to 3.2-fold with respect to the control group (Figure 4). Most Gal3^+^ cells are Iba1^+^. Therefore, we can conclude that expression of Gal3 increases in liver macrophages upon LPS administration (Figure 5).This analysis supports the view that tissue-specific Gal3 expression may play a critical role in driving LPS-associated dysregulated immune responses.

Since Gal3 is a known ligand of TLR4, we also wanted to know the effect that deletion on *Gal3* has on the expression levels of TLR4 mRNA after LPS injection. Our PCR analysis showed that TLR4 mRNA levels increased significantly in macrophages from WT animals 1 h after LPS injection. This effect was prevented in Gal3KO animals (Figure 3G). The reduction in TLR4 mRNA levels in animals lacking *Gal3* could explain the better outcome of the Gal3KO mice after the LPS challenge. TLR4 mRNA levels in liver and spleen were very low in control animals, and these levels were not affected by LPS injection (Figure 3E,F).

### 2.3. Genetic Deletion of Gal3 Alters the Immune Response in the Liver, Spleen and Peritoneal Macrophages

As already stated, deregulation of the immune system may lead to cytokine storms in which both pro- and anti-inflammatory mediators are involved. Therefore, in order to study the inflammatory events in our experimental conditions, we proceeded to quantify by qPCR the expression of *TNF-α*, *iNOS*, *IL-1β*, and *IL-6* as pro-inflammatory markers, and *YM1*, *IL-10*, and *arginase* as anti-inflammatory markers, in the biological samples extracted (liver, spleen, and peripheral macrophages). Our results showed that in the liver, LPS injection in WT animals increased the mRNA levels of the pro-inflammatory markers *TNF-α*, *iNOS*, *IL-1β*, and *IL-6* (Figure 6A–D). Moreover, LPS treatment also increased the mRNA expression levels of the anti-inflammatory markers *IL-10* and *arginase* (Figure 6F,G).

Interestingly, genetic ablation of *Gal3* decreased the mRNA levels of *iNOS* in LPS-treated animals (Figure 6B) by about 60%. However, at this time point, mRNA levels of *TNF-α*, *IL-1β*, and *IL-6* of Gal3KO LPS-treated animals increased nearly 3, 2, and 2.5 fold respectively, with respect to the WTLPS group (Figure 6A,C,D). Regarding the anti-inflammatory markers, Gal3KO mice showed an increase in the mRNA levels of *YM1* and *IL-10* (6 and 2 fold respectively, with respect to the WTLPS group; Figure 6E,F), but a decrease in mRNA *arginase* levels (more than half with respect to WTLPS group; Figure 6G).

In the spleen, LPS treatment in WT animals induced an increase in the mRNA levels of *TNF-α*, *IL-1β*, and *IL-6* (Figure 6H–K). LPS treatment also induced an increase in the mRNA levels of *IL-10* (Figure 6M). Genetic deletion of *Gal3* induced a higher increase in mRNA levels of *TNF-α* and *IL-1β* (Figure 6H,J) and an increase in the levels of *YM1* and *arginase* (Figure 6L,N), but a decrease in the mRNA levels of *IL-10* (Figure 6M). No statistical differences were found in *iNOS* mRNA levels (Figure 6I).

Finally, in peripheral macrophages, LPS treatment in WT animals increased the expression levels of *TNF-α*, *iNOS*, *IL-1β*, and *IL-6* (Figure 6O–R). Levels of anti-inflammatory markers, such as *YM1*, *IL-10*, and *arginase* also increased in WTLPS mice (Figure 6S–U). In this case, genetic deletion of *Gal3* decreased the mRNA expression levels of *iNOS* and *arginase* (Figure 6P,U).

Since this PCR analysis did not allow us to draw conclusions about the effect that *Gal3* has on the immune response in the tissue, we next studied the effect of LPS injection on immune cell infiltration. Therefore, we performed immunohistochemistry analysis against CD68 (a marker of macrophages) and CD4 (a marker of lymphocytes). Infiltration of lymphocytes and macrophages increased in the liver 36 h after LPS injection. Interestingly, this effect was effectively abolished in the Gal3KO animals (Figure 7 and Figure 8).

### 2.4. Absence of Gal3 Reduces Serum Levels of TNF-α and IL-6 in Response to LPS Challenge

In addition to the overexpression of genes directly related to inflammation, macrophages release pro-inflammatory cytokines to the extracellular space. Therefore, we also wanted to know whether the absence of *Gal3* could reduce the protein levels of TNF-α and IL-6 in serum. Using an ELISA kit, we found that the levels of TNF-α and IL-6 increased significantly in response to LPS challenge (around 275 and 62 fold with respect to the control group (WT), respectively) (Figure 9A,B, *p* < 0.001). This increase was, however, prevented in Gal3KO mice, demonstrating the important role that Gal3 seems to play in the immune response (cytokine storm) that takes place following our model of LPS-induced endotoxemia.

### 2.5. Gal3 Depletion Protects from Tissular Damage

Sections from different organs were analyzed according to the classical procedures of pathological anatomy and a histological score was performed from each organ analyzed. Consequently, we have analyzed different organs that suffer histological damage under LPS-induced endotoxemia conditions, including liver (Figure 10 and Appendix A), spleen (Figure 11 and Appendix A), and lung (Figure 12 and Appendix A) using optical and electron microscopies. The WTLPS group showed serious alterations in these organs (Figure 10C,D, Figure 11C,D and Figure 12C,D and Appendix A). For instance, the existence of a hyperplasia and hypertrophy of the lymphoid follicles were easily distinguished in these animals (Figure 11C and Appendix A). The red splenic pulp was apparently normal. Ultrastructural studies showed that only lymphocytes, lymphoblasts, and reticular cells were featured (Figure 11D and Appendix A). The liver of WTLPS animals showed some alterations, although the normal structure of this organ was maintained. Most hepatocytes of the lobule, especially those of the perilobular space, showed steatosis preferentially multilocular (Figure 10C). In general, the pulmonary parenchyma showed a generalized atelectasis, both at the bronchial and alveolar levels (Figure 12C and Appendix A). With the electronic microscope, we found thickenings of its septal structure, with large numbers of pneumocytes II and macrophages (Figure 12D and Appendix A).

The absence of *Gal3* protected against damage in the liver (Figure 10E,F and Appendix A), spleen (Figure 11E,F and Appendix A), and lung (Figure 12E,F and Appendix A). Hence, in the Gal3KOLPS group, the lesions were scarce, although the hypertrophy and hyperplasia of the lymphoid follicles of the spleen were maintained (Figure 11E and Appendix A). The liver maintained its structure and showed little hepatic steatosis (Figure 10E). Moreover, the lung maintained its bronchial and alveolar composition organs (Figure 12E).

We also analyzed the effect of LPS-induced endotoxemia in the brain and the effect of genetic deletion of *Gal3* in these conditions (Figure 13A–G and Appendix A). Intriguingly, most important lesions after LPS treatment were found in the cerebral cortex. Both optical and electron microscopies and immunohistochemistry against Iba1 showed an increase and mobilization of glia cells, diffuse and in nodulations (Figure 13C,D,H,I). Microglia also showed the presence of lysosomal components similar to lipofuscin (Appendix A). In Gal3KOLPS animals, although there was an increase in glial cells, there were scarcely any neuronal modifications (Figure 13E,F and Appendix A).

## 3. Discussion

In this study, we have analyzed the role of Gal3 during endotoxemia, a condition that causes over-activation of the immune system, thus leading to a pro-inflammatory response. Importantly, LPS-induced endotoxemia is often used to model the acute inflammatory response associated with early sepsis [6]. Indeed, it is known that in the pathogenesis of LPS-induced endotoxemia, SIRS and coagulation disturbance play a pivotal role, finally leading to disseminated intravascular coagulation (DIC) and multiple organ dysfunction syndrome (MODS). Under these conditions, clinical manifestations such as fever, hypothermia, tachypnea, tachycardia, leukocytosis, and leukopenia, among others, are evident. Molecular events include activation of TLR4 with the subsequent expression and release of pro-inflammatory cytokines and upregulation of tissue factor and coagulation-related proteases to further lead to DIC along with impairment of blood flow [33]. These pathological events are critically associated with organ failure [34], hence the need to find effective therapies to treat the excessive and uncontrolled systemic inflammatory response [35,36].We provide evidence that the absence of *Gal3* reduces the toxicity in a model of LPS-induced endotoxemia as evidenced by a significant reduction in the mortality rate and more preserved tissue integrity. Our study may shed light onGal3-related functions under conditions of endotoxemia and sepsis.

Gal3 is strongly expressed in myeloid cells including monocytes, macrophages, DCs, and neutrophils [37]. Subcellular localization of Gal3 includes cytoplasm, nucleus, and membrane [37,38]. Interestingly, membrane-associated Gal3 plays immune-associated roles under acute and chronic inflammation [37]. We took advantage of flow cytometry to measure the amount of membrane-bound Gal3 in most blood immune cells after LPS challenge, including monocytes, neutrophils, DCs, and different subsets of T cells and B cells. Myeloid cells showed much higher levels of membrane-bound Gal3, especially monocytes. As a rule, after systemic LPS challenge, all immune cells lacked the ability to bind Gal3 at the membrane surface. This finding supports the notion that Gal3 may act as an alarmin—i.e., a molecule that, when released by immune cells, is able to induce a sterile immune or inflammatory response [39,40]. We measured serum levels of Gal3 under control and LPS conditions and found, unexpectedly, a significant decrease under conditions of endotoxemia, a clear indication that Gal3 is either internalized by blood immune cells, degraded or extravasated to peripheral organs. Indeed, since we found a complete lack of Gal3 binding by circulating immune cells, the possibility that Gal3 may thus spread to different organs to orchestrate immune-associated functions is certainly plausible. Gal3 has been found to promote leukocyte adhesion to the endothelium and slow rolling, and recruitment of neutrophils and monocytes [41,42]. Consequently, an efficient recruitment of leukocytes during acute inflammatory response may rely on Gal3 recruitment to target tissues. This assumption would require the unequivocal presence of highly reactive Gal3 in peripheral organs. Thus, we analyzed the expression of Gal3 in liver, spleen, and lung 36 h after LPS challenge. Our immunohistochemical analysis revealed a robust up-regulation of Gal3 in WT but not in Gal3KO animals in all analyzed organs, thus sustaining an important role of this galectin in driving immune-related responses under conditions of endotoxemia.

To delve into the modulation of immune cells by Gal3 at the molecular level, the expression levels of TRL4, a Gal3 ligand, was measured 1 h after LPS injection in WT and Gal3KO mice. This analysis demonstrates a nice effect of *Gal3* deletion on *TLR4* expression in macrophages as it prevented the LPS-induced expression of this receptor (Figure 3). It is known that TLR4 is one of the main drivers of microglia activation [43], triggering several transduction pathways, such as the nuclear factor kappa B (NF-κB) pathway and the mitogen-activated protein kinases (MAPKs) pathway, which cause increased expression of inflammatory cytokines. Therefore, our results show that the effects on immune responses driven by Gal3 could be mediated by TLR4 and its downstream signaling cascade.

Studies aimed at defining the role of Gal3 under conditions of sepsis and endotoxemia have led to opposing conclusions. In a previous study, Li et al. reported that Gal3 acts as a negative regulator of LPS-induced endotoxemia and inflammation, in sharp contrast to our study [29]. According to their study, animals lacking *Gal3* succumbed earlier to systemic LPS and exhibited higher serum TNF-α levels after LPS challenge. In the same study, *Gal3* deletion exacerbated the LPS-induced inflammatory response of bone marrow-derived macrophages [29]. The authors found that Gal3 binds to LPS, thus inhibiting the LPS-associated pro-inflammatory response as tested in bone marrow-derived macrophages [29]. A plausible interpretationof this study would be that Gal3 represses LPS function (i.e., by interfering with the interaction of LPS with its receptor complex). However, the functional relevance of the interaction between Gal3 and LPS in vitro is far from clear [31,32]. It is intriguing that studies reporting potential interaction between Gal3 and LPS have led to opposite effects, including enhancement of the pro-inflammatory activity of LPS [44] and inhibition of the pro-inflammatory activity of LPS [29]. Indeed, our qPCR analysis was not enough for drawing any conclusions about the pro-inflammatory role of Gal3 in our model of endotoxemia in the different organs analyzed. A plausible explanation for this apparent discrepancy is related to the fact that time-associated cytokine changes are very fast and transient [45]. Interestingly, increased circulating levels of TNF-α and IL-6 are considered the main features of the inflammatory response associated with sepsis [2] and play a key role in the pathophysiology of severe sepsis [46]. In fact, among the cytokines whose levels are increased during sepsis, IL-6 is the one that shows a better correlation with the mortality rate [47]. Consequently, we measured the circulating levels of both pro-inflammatory cytokines in our experimental animals. Our study demonstrates that, in the absence of *Gal3*, the early LPS-induced peak of circulating TNF-α and IL-6 is significantly attenuated, thus further supporting the deleterious effect of Gal3 under conditions of endotoxemia. Our previous studies on microglial cells revealed a clear synergistic effect of Gal3 on the LPS-induced pro-inflammatory response [7,20]. Thus, we demonstrated that activated microglia release Gal3, which plays an essential role for the full microglial response upon LPS stimulation [7]. In previously published papers, our group has used two experimental approaches to inhibit Gal3 in vitro (BV2 microglial cell line): *Gal3* expression was suppressed using siRNA and Gal3 blocking antibody was used to neutralize the effects of the released Gal3. In this study, we observed that both methods prevented the LPS-induced inflammation. Moreover, to validate the Gal3 effect over the inflammatory response upon LPS stimulus, the release of several proinflammatory cytokines were checked in WT and Gal3KO primary cell cultures, confirming the BV2 cell data, with reduced inflammatory response in Gal3KO microglia [7]. Indeed, *Gal3* inhibition by either gene silencing or using a neutralizing antibody repressed the LPS-induced pro-inflammatory response of microglia, and importantly, we demonstrated that Gal3 acts as an endogenous true ligand of TLR4 [7]. Quite different studies support the view that Gal3 acts as a powerful pro-inflammatory signal including activation of the NADPH oxidase system [48] and stimulation of superoxide production from neutrophils [49]. Gene deletion of *Gal3* suppresses production of pro-inflammatory cytokines and reduces NLRP3 inflammasome activation, the molecular platform leading to caspase-1 activation and subsequent cleavage and release of IL-1β [50]. Further, Gal3 has been shown to amplify inflammation of atherosclerotic plaque progression through macrophage activation and monocyte chemoattraction [51]. Overall, we may conclude that data supporting a pro-inflammatory role of Gal3 areoverwhelming. A plausible explanation for the contrasting effects of Gal3 on LPS-associated endotoxemia found by Li et al. (2008) and our study may rely on the LPS doses used in each study. The lethal dose of 50 (LD50) LPS is about 1–25 mg/kg [2]. While Li et al. used an LPS dose of 23 mg/kg (upper limit), we used a dose of 5 mg/kg (lower limit). These differences in LPS doses may certainly be behind the opposite effects (beneficial vs detrimental) ascribed to Gal3 under conditions of endotoxemia. Serum levels of Gal3 increase in patients suffering from sepsis [25,27,52] and in experimental animal models of sepsis such as that induced by CLP [27]. In fact, Gal3 is useful as a biomarker for the prediction of mortality in sepsis [25,53]. A detrimental role of Gal3 is deduced from Gal3KO mice experiments suffering CLP-induced sepsis, which are more resistant than WT mice [27]. In a different model of sepsis induced by infection with Francisellanovicida, the absence of *Gal3* led to significantly longer survival time, and showed reductions in levels of well-established sepsis mediators including vascular injury markers, inflammatory cytokines, and acute phase proteins [28]. The amplifying pro-inflammatory nature of Gal3 was evaluated in vitro in bone marrow-derived macrophages and peritoneal neutrophils in response to Francisellanovicida [28].

To demonstrate conclusively that Gal3 plays a deleterious role in SIRS, and considering that appearance of MODS is critical for life expectancy of patients, we took advantage of (i) optic and electronic microscopy techniques to analyze those organs severely compromised during SIRS including liver, spleen, and lung and (ii) analysis of the death rate. Our microscopy analysis demonstrated that the absence of *Gal3* preserved tissue integrity in all analyzed organs. Interestingly, we also found that LPS-induced endotoxemia produced significant brain damage, especially in the cortex, with strong glial mobilization. It has been demonstrated that cytokine production is an early event after sepsis, participating in blood–brain barrier alterations and brain damage [54]. The absence of *Gal3*, however, prevented these alterations in all the organs examined. Indeed, most solid evidence sustaining a detrimental role of Gal3 triggered by LPS-induced endotoxemia relies on the delayed death rate ascribed to Gal3KO mice. The absence of *Gal3* significantly decreased the mortality rate following LPS treatment as compared with WT animals. Early treatment of sepsis is necessaryto minimize fatal outcomes. Our study supports the usage of inhibitors of Gal3 to increase the therapeutic window in which other treatments could be administered.

## 4. Materials and Methods

### 4.1. Animals and Treatments

12–15-week-old male C57BL/6 (wild-type, WT) and Gal3 null mutant (Gal3KO) mice (20–25 g) were obtained from the Center of Production and Animal Experimentation (Espartinas, Seville, Spain). They were housed at a constant room temperature of 22 ± 1 °C and relative humidity (60%), with a 12-h light-dark cycle and ad libitum access to food and water.

Animals were distributed within 4 groups according to two different variables: solution administrated intraperitoneally (saline or LPS in 0.9% sterile saline) and genotype (WT or Gal3KO). In the WT group (the control group), WT animals received an intraperitoneal injection of saline; in the KO group, Gal3KO animals received an intraperitoneal injection of saline; in the WTLPS group, WT animals received an intraperitoneal injection of LPS (from Escherichia coli serotype 0111: B4; Sigma Aldrich, St. Louis, MO, USA); finally, in the KOLPS group, Gal3KO animals received an intraperitoneal injection of LPS. All treatments were administrated in a volume of 50 µL per 25 g of body weight.

Animals were sacrificed at different time points depending on the technique assayed. These time points were chosen based on previous studies [45,55]. For qPCR, enzyme-linked immunosorbent assay (ELISA), and flow cytometry assays, animals were sacrificed 1 h after the saline/LPS injection, time in which inflammatory markers are high according to the literature [56,57]. For histological studies, animals were sacrificed 36 h after the injection. At least 3 animals per group were used, except for the survival study in which 12 animals were used per group. Whenever possible, peritoneal macrophages, blood, and organs were obtained from the same animal (Figure 14).

### 4.2. Survival Analysis

In order to select the optimal LPS dose to be used, a dose-response study using 5 (Appendix A), 7.5 (Appendix A), 10 (Appendix A), and 15 mg/kg (Appendix A) was performed to analyze survival following LPS challenge. To assess the effect of this treatment and verify the appearance of typical alterations of endotoxemia, animals were observed every 6 h for general health. The symptomatology was evaluated for 80 h and scored using a scale (Table 1) based on our observations and the criteria proposed by Benavides et al. [58]. From this analysis, a dose of 5 mg/kg LPS was selected, in accordance with previous studies [45,59].

### 4.3. Flow Cytometry

Mice were anesthetized and blood was collected from the heart. To obtain peripheral mononuclear cells (PBMCs), blood was collected and placed in a cold ammonium chloride lysis solution (82.8 g/L NH_4_CL, 10.1 g/L KHCO_3_, 0.37 g/L EDTA (MERCK, Darmstadt, Germany)) for 10 min. Then, cellular suspension was washed twice with a cold phosphate-buffered saline (PBS).

Then, PBMCs were stained by surface marker antibodies for 20 min at 4 °C. Antibodies used were: anti-Ly6G FITC (REA526), anti-CD3e VioBlue (145-2C11), anti-Ly6C PE (1G7.G10), anti-CD45 APC (30F11) (Miltenyi Biotec, Bergisch Gladbach, Germany), anti-CD11c PerCPCy5.5 (N418), anti-MHC-II SB600 (M5/114.15.2), CD4 AF700 (GK1.5), CD11b PE-eFluor610 (M1/70), CD19 SB702 (eBio1D3) (eBioscience, San Diego, CA, USA), CD8 APC-eFluor 780 (53–6.7) (Invitrogen, Waltham, MA, USA), and anti-Gal3 PECy7 (M3/38) (Biolegend, San Diego, CA, USA). Dead cells were excluded using LIVE/DEAD fixable Aqua Blue Dead Cell Stain (Life Technologies, Carlsbad, CA, USA). Gal3 expression was analyzed in B-cells (CD19^+^CD3e^−^), CD4^+^ T-cells (CD3e^+^CD4^+^CD8^−^), CD8^+^ T-cells (CD3e^+^CD4^−^CD8^+^), double negative T-cells (CD3e^+^CD4^+^CD8^−^), dendritic cells (DCs) (CD3e^−^CD19^−^CD11c^+^MHC-II^+^), neutrophils (CD3e^−^CD19^−^CD11c^+^CD11b^+^Ly6G^+^Ly6Cdim), monocytes (CD3e^−^CD19^−^CD11c^+^CD11b^+^Ly6G-Ly6Chigh), and macrophages (CD3e^−^CD19^−^CD11c^+^CD11b^+^Ly6G-Ly6C^−^). A representative gating strategy is shown in Appendix A. Fluorescence Minus One (FMO) plus isotype antibody controls were used to determine the expression of Gal3 in each cellular subset of stimulated with LPS and unstimulated mice (Appendix A). A minimum of 500,000 events were acquired. Flow cytometry was performed on a BD LSR Fortessa (BD Bioscience, Mississauga, ON, Canada). Analysis was performed using FlowJo version 9.2 (Tree Star, Ashland, OR, USA) and data are expressed as frequencies (%).

### 4.4. Isolation and Culture of Primary Peritoneal Macrophages

Peritoneal macrophages were isolated from mice from different experimental groups according to the protocol developed by Davies [60]. First, animals were anesthetized with isoflurane. Subsequently, 10 mLof a 1× PBS solution was injected intraperitoneally, and the entire body was shaken for 10 s. Then, PBS-containing resident peritoneal cells were withdrawn, centrifuged (400× *g* 5 min), and resident peritoneal cells were resuspended in an RPMI-1640 medium (Sigma-Aldrich, San Luis, MO, USA) supplemented with 10% fetal bovine serum (Sigma-Aldrich, San Luis, MO, USA) containing 50 IU of penicillin, 50 µg streptomycin, and 2 mM glutamine per milliliter (Gibco, Invitrogen Ltd., Waltham, MA, USA). Three hundred thousand cells per well were plated in 24-well plates and incubated for 60 min at 37 °C. Then, the non-adherent cells were removed by being washedfive times with PBS.

### 4.5. RNA Extraction and cDNA Synthesis

Liver and spleen were extracted 1 h after different treatments and stored at −80 °C until use. Total RNA was isolated using the RNeasy Mini Kit (Qiagen, Düsseldorf, Germany) according to the manufacturer’s recommendations. The samples were previously disrupted by the use of a TissueLyser II (Qiagen, Düsseldorf, Germany). On the other hand, total RNA from peritoneal macrophages was isolated using TRIsure™ (Bioline, London, UK), following the instructions established by the manufacturer. In both cases, the amount of total RNA obtained in each sample was measured spectrophotometrically using the NanodropTM 2000/2000c (Thermo Fisher Scientific, Waltham, MA, USA). The volume required of each sample was calculated to perform reverse transcription of 1 μg of RNA using the RevertAid™ First Strand cDNA kit Synthesis Kit (Thermo Fisher Scientific, Waltham, MA, USA). The RT-PCR was performed in a PTC-100 thermocycler (MJ research Inc., Waltham, MA, USA) using a program consisting of 5 min at 25 °C, then 60 min at 42 °C and, finally, 5 min at 70 °C. The resulting cDNAs were stored at −80 °C until use.

### 4.6. Real Time PCR

Real-time PCR was performed with SensiFASTTM SYBR No-ROX Kit (Bioline, London, UK), 0.4 μM primers, and 4 μg cDNA. Controls were carried out without cDNA. Amplification was run in a LightCycler 480 (Roche Molecular Systems, Pleasanton, CA, USA) thermal cycler at 95 °C for 2 min followed by 40 cycles of 95 °C for 5 s, 65 °C for 10 s, and 72 °C for 20 s, and ended with a cycle of 7 min at 72 °C. Following amplification, a melting curve analysis was performed by heating the reactions from 65 to 95 °C in 0.1 °C/s while monitoring fluorescence. Analysis confirmed a single PCR product at the melting temperature. *β-actin* served as the reference gene and was used for sample normalization. Primer sequences for *β-actin*, inducible nitric oxide synthase (*iNOS*), *TNF-α*, *IL-1β*, *IL-6*, *Gal3*, *arginase*, *YM1* (chitinase-like 3), and *TLR4* are shown in Table 2. The cycle at which each sample crossed a fluorescence threshold (Ct) was determined, and the triplicate values for each cDNA were averaged. Analyses of real-time PCR were done using a 2Ct relative quantification method [61].

### 4.7. ELISA

Fresh peripheral blood was collected from animals that had receivedthe different treatments in vacuum filled tubes and incubated at room temperature for 30 min to allow coagulation. Then, the blood was centrifuged at 2000× *g* for 15 min at 4 °C to obtain serum, which was stored at −40 °C until used. Serum Gal3, TNF-α, and IL-6 concentrations were determined by using a Mouse Galectin-3 Duo Set ELISA DY1197-05, a Mouse TNF alpha ELISA Ready-SET-GO! ^®^, and a Mouse IL-6 ELISA Ready-SET-GO! ^®^ kit (Affymetrix, Santa Clara, CA, USA), respectively, following the manufacturer’s instructions. The plates were read on a Synergy HT multimodal plate reader (BioTek, Winooski, VT, USA) set to 450 nm. All conditions were assayed in duplicate.

### 4.8. Immunohistological Evaluation of Gal3, CD4 and CD68

Thaw-mounted 20-μm liver, lung, and spleen sections were cut on a cryostat at −15 °C and mounted in gelatin-coated slides. Primary antibodies used were goat-derived anti-Gal3 (R&D Systems, Minneapolis, MN, USA; 1:100), rat-derived anti-CD4 (Santa Cruz Biotechnologies, Santa Cruz, CA, USA; 1:100), and mouse-derived anti-CD68 (Invitrogen, Waltham, MA, USA; 1:100). Incubations and washes were conducted in Tris-buffered saline (TBS), pH 7.4. All work was performedat room temperature. Sections were washed and then treated with 0.3% hydrogen peroxide in methanol for 20 min, washed again, and incubated in a solution containing TBS and 1% horse/goat serum (Vector, Burlingame, CA, USA) for 60 min in a humid chamber. Slides were drained and further incubated with the primary antibody in TBS containing 1% horse/goat serum and 0.25% TritonX-100 for 24 h. Sections were then incubated for 2 h with biotinylated horse anti-goat, goat anti-mouse, or goat anti-rat IgG (Vector, Burlingame, CA, USA; 1:200). The secondary antibody was diluted in TBS containing 0.25% TritonX-100, and its addition was preceded by three 10-min rinses in TBS. Sections were then incubated with VECTASTAIN^®^Elite^®^ABC Kit, peroxidase (Vector, Burlingame, CA, USA). The peroxidase was visualized with a standard diaminobenzidine/hydrogen reaction for 5 min.

### 4.9. Immunohistochemistry Data Analysis

Analysis was performedusing the AnalySIS imaging software (Soft Imaging System GmbH, Münster, Germany) coupled witha Polaroid DMC camera (Polaroid, Cambridge, MA, USA) attached to a Leika light microscope (LeikaMikroskopie, Wetzlar, Germany). For counting cells showing Gal3, CD4, and CD68 immunoreactivity, a systematic sampling of the area occupied by positive cells for these markers was made from a random starting point with a grid adjusted to count five fields per section. An unbiased counting frame of the known area (40 × 25 µm = 1000 µm^2^) was superimposed on the tissue section image under a 40× objective. Cells showing Gal3, CD4, or CD68 immunoreactivity were counted using five fields per section and three sections per animal and the number of cells was expressed as cells per mm^2^.

### 4.10. Immunofluorescence of Iba-1 and Gal3

Animals were perfused and brain sections were prepared as described above. Incubations and washes for all the antibodies were conducted in PBS, pH 7.4. All work was performedat room temperature, unless otherwise noted. Sections were blocked with PBS containing 5% BSA for 2 h. The slides were then incubated overnight at 4 °C with the primary antibodies: rabbit-derived anti-Iba1 (Wako, Osaka, Japan; 1:500) and goat-derived anti-Gal3 (R&D Systems, Minneapolis, Canada, USA; 1:200). Primary antibodies were diluted in PBS containing 1% BSA and 1% Triton X-100. After three washes in PBS, sections were incubated with secondary antibodies conjugated to Alexa Fluor^®^ 488 (Invitrogen, Waltham, MA, USA; 1:500) and Alexa Fluor^®^ 647 (Invitrogen, Waltham, MA, USA; 1:500) for 2 h at room temperature in the dark. Fluorescence images were acquired using a confocal laser scanning microscope (Zeiss LSM 7 DUO, Oberkochen, Germany) and processed using the associated software package (ZEN 2010, Oberkochen, Germany).

### 4.11. Light and Electron Microscopy

Tissue samples for histological examination from the liver, spleen, lung, and brain were taken 36 h after the saline/LPS injection. For light microscopy, samples were first fixed in a 10% buffered formalin for 24 h at 4 °C, and then immediately dehydrated in graded series of ethanol, immersed in xylol, and embedded in paraffin wax using an automatic processor. Sections of 3–5 µm were mounted. After they had been deparaffinized, the sections were rehydrated, stained with hematoxylin and eosin, and mounted with Cristal/Mount [62]. A semiquantitative evaluation of the histological damage was made, including hypertrophy and hyperplasia of lymphoid follicles, steatosis, atelectasis, glial cell mobilizations, and catarrhal steatosis.

For electron microscopy (MET), the samples were prefixed in 2% glutaraldehyde fixative (in pH 7.4 phosphate buffer for 10 h at 4 °C) and postfixed in 1% osmium tetroxide fixative (in pH 7.4 phosphate buffer for 0.5 h at 4 °C). Subsequently, they were dehydrated in graded ethanol series and embedded in EPON. Ultra-thin sections, 50–60 nm, were cut with an LKB microtome. The sections were mounted on a copper grid and stained with uranylacetate and lead citrate. The tissue sections were examined in a JEOL JEM 1400 MET (Tokyo, Japan) [62].

For the evaluation of the tissue damage, we rely on an observation of the histopathological sections under an optical microscope, which is verified by observations under an electronic microscope. In addition, a semi-quantitative evaluation of the observations was conducted. Histopathological evaluation is performed in a blind manner by two highly experienced pathologists.

### 4.12. Statistical Analysis

The sample size was calculated with the ENE 3.0 software (GlaxoSmithKline, Madrid, Spain). Animals were randomly distributed so that each box had different ages and weights. Results are expressed as mean ± SD. Means were compared by one-way ANOVA followed by the Fisher’s LSD test for post hoc multiple range comparisons, unless otherwise noted. For flow cytometry, Prism, version 5.0 (GraphPad Software Inc., San Diego, CA, USA) was used for the generation of the graphs [63]. Statistical significance analysis was calculated using the Mann–Whitney U test. Results are expressed as median ± IQR. For Gal3 ELISA and histological studies, a two tailed Student’s *t* test was used. The comparison of the survival curves was performed with the Kaplan–Meier method; specifically, the Log-Rank test was used. An alpha level of 0.05 was used. The Statgraphics 18 × 64 statistical package and the Statistical Package for the Social Sciences software (SPSS 21.0, New York, NY, USA), were used for the analyses.

## Figures and Tables

**Figure 1 ijms-23-01170-f001:**
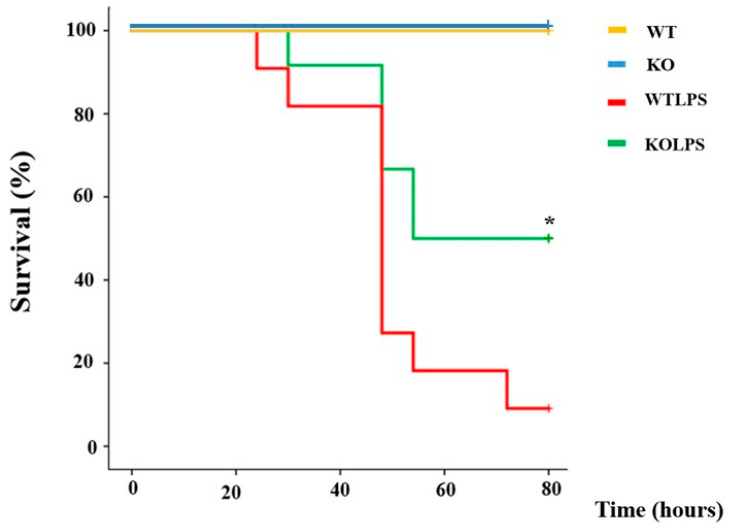
Gal3 knockout mice are more resistant to septic shock. WT and Gal3KO mice were subjected to an intraperitoneal injection of LPS (5 mg/kg of body weight) or saline solution (*N* = 12 animals per group). The mortality rate was monitored regularly for 80 h and represented as percentage of survival. The statistical analysis was performed using the Log-Rank Test. Abbreviations: WT, wild type mice; KO, Gal3 knockout mice; WTLPS, wild type mice treated with LPS; KOLPS, Gal3 knockout mice treated with LPS. *, *p* < 0.05.

**Figure 2 ijms-23-01170-f002:**
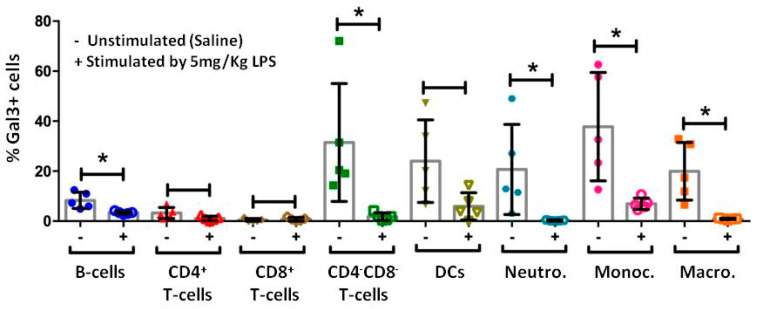
Determination of Gal3 on immune cells of peripheral blood. Levels of Gal3 were determined by flow cytometry in peripheral blood of mice (*N* = 5 animals). Briefly, after blood collection, erythrocytes were lysed using an ammonium chloride lysis solution. Cells were washed and stained with surface marker antibodies for 20 min on ice. Gal-3 expression was analyzed in B-cells, CD4^+^ T-cells, CD8^+^ T-cells, double negative T-cells, dendritic cells, neutrophils, and macrophages. Results are expressed as median ± IQR. Statistical significance was calculated using the Mann–Whitney U test. Abbreviations: DCs, dendritic cells; Macro, macrophages; Monoc, monocytes; Neutro, neutrophils. *, *p* < 0.01.

**Figure 3 ijms-23-01170-f003:**
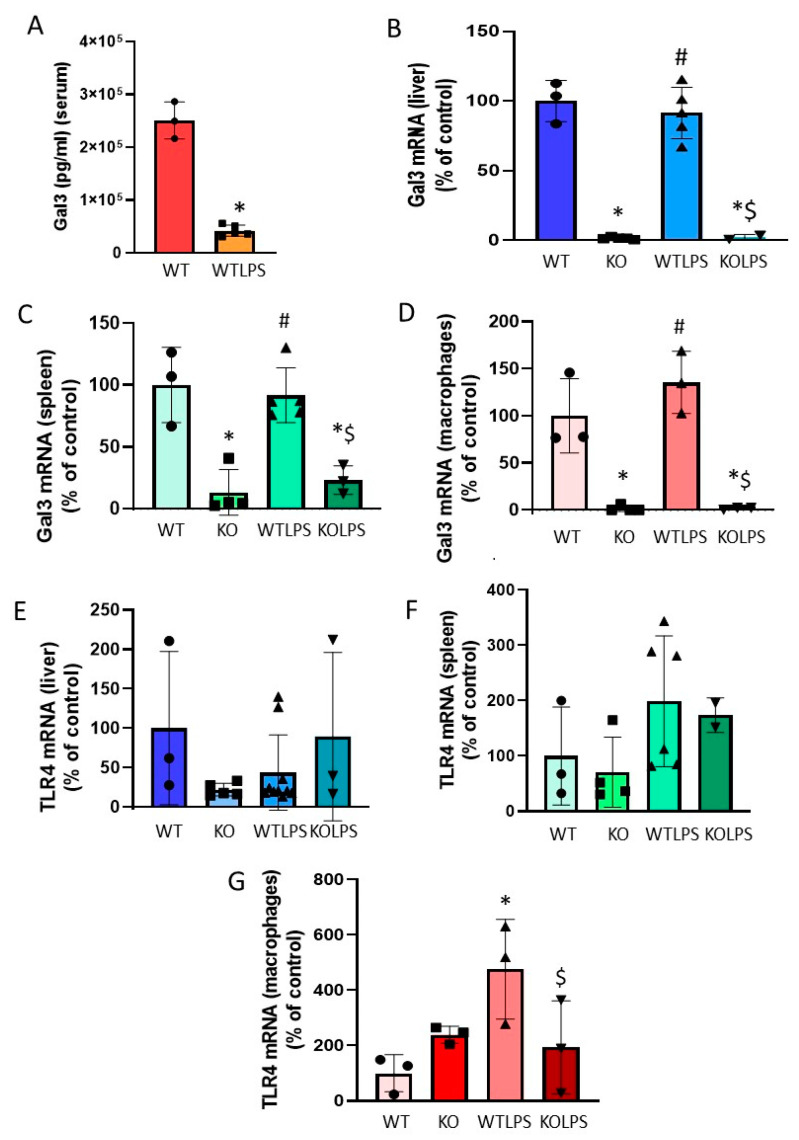
Determination of Gal3 and TLR4. (**A**)The expression of Gal3 in serum was measured by ELISA. Blood samples were collected from the heart of mice 1 h after LPS/saline injection. Results are mean ± SD of *N* = 4–6 animals, expressed as ng/mLof the analyzed protein and relative to the WT group. Statistical significance (two tailed Student’s *t* test): *p* < 0.001. Using RT-PCR, the mRNA expression of Gal3 was measured in the liver (**B**), spleen (**C**), and peritoneal macrophages (**D**). Using RT-PCR, the mRNA expression of TLR4 was measured in the liver (**E**), spleen (**F**), and peritoneal macrophages (**G**). For PCR analysis, animals were culled 1 h after LPS/saline injection. Results are mean ± SD of *N* = 3–10 animals, normalized to *β-actin* and expressed as relative expression to the WT group. Statistical significance (one-way ANOVA followed by the LSD post hoc test for multiple comparisons): *—compared with WT group; #—compared with KO group; $—compared with WTLPS group; *p* < 0.001 for (**B**–**D**), *p* < 0.05 for (**G**). Abbreviations: WT, wild type mice; KO, Gal3 knockout mice; WTLPS, wild type mice treated with LPS; KOLPS, Gal3 knockout mice treated with LPS.

**Figure 4 ijms-23-01170-f004:**
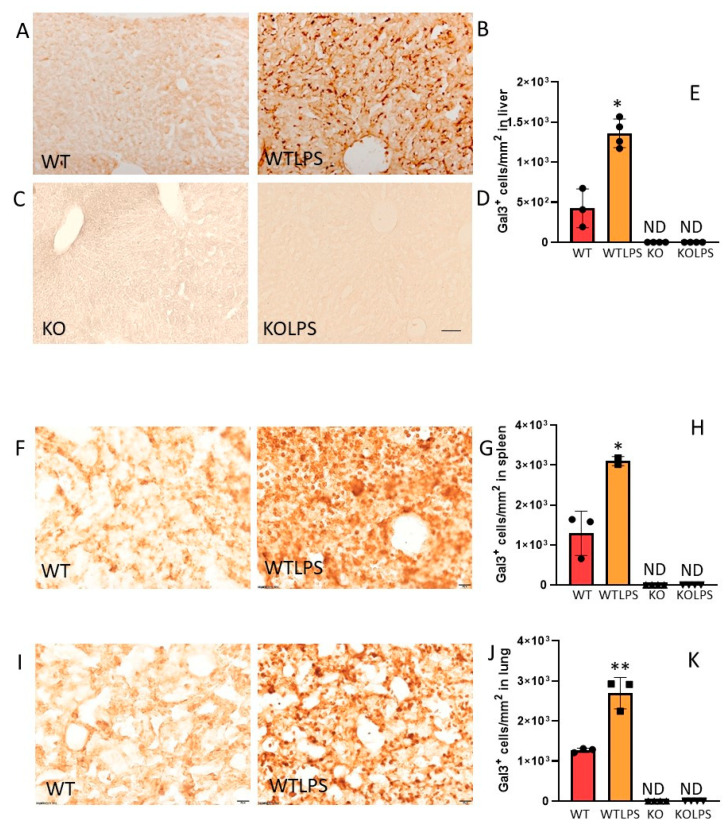
Gal3 expression in the liver, lung, and spleen of mice from the different treatments assayed. (**A**) Section from liver showing some Gal3 positive cells in a WT animal. (**B**) Immunoreactivity of Gal3 in a liver section from a WT animal injected i.p. with LPS. (**F**) Section from spleen showing some Gal3 positive cells in a WT animal. (**G**) Immunoreactivity of Gal3 in a spleen section from a WT animal injected i.p. with LPS. Again, a strong reaction can be seen. (**I**) Section from lung showing some Gal3 positive cells in a WT animal. (**G**) Immunoreactivity of Gal3 in a lung section from a WT animal injected i.p. with LPS, showing a strong reaction. As expected, in the case of Gal3KO animals, immunoreactivity of Gal3 was not found in any treatment assayed (**C**,**D**). Scale bar: (**A**–**D**), 100 μm; (**F**–**J**), 20 μm. Quantification of the density of Gal3 positive cells in liver (**E**), spleen (**H**), and lung (**K**) from the different treatments assayed. Results are mean ± SD of *N* = 3–4 animals, expressed as number of cells per mm^2^. Statistical significance (two tailed Student-*t* test): * *p* < 0.01; ** *p* < 0.05. Abbreviations: WT, wild type mice; WTLPS, wild type mice treated with LPS; KO, Gal3 knockout mice; KOLPS, Gal3 knockout mice treated with LPS.

**Figure 5 ijms-23-01170-f005:**
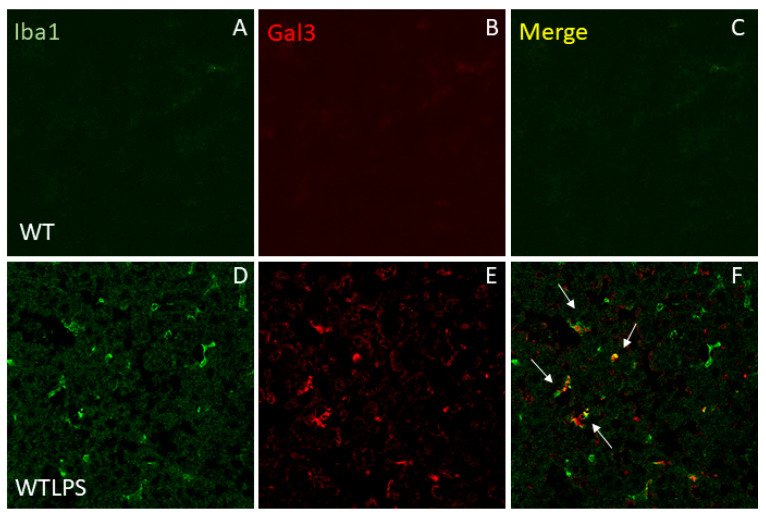
Double immunofluorescence of Iba1 andGal3 expression. (**A**) Iba1 staining in the liver of a WT animal. Virtually, no monocytes/macrophages are infiltrated in control animals. (**B**) Section from liver showing some Gal3 positive cells in a WT animal. (**C**) Merge image showing co-localization of Iba1^+^ cells and Gal3^+^ cells. (**D**) Immunoreactivity of Iba1 in a liver section from a WT animal injected i.p. with LPS. A strong reaction can be seen. (**E**) Immunoreactivity of Gal3 in WT animals injected with LPS. (**F**) Merge image showing co-localization of Iba1^+^ cells and Gal3^+^ cells (arrows). Most Gal3^+^ cells co-localize with Iba1^+^ cells. Scale bar: 50 μm.

**Figure 6 ijms-23-01170-f006:**
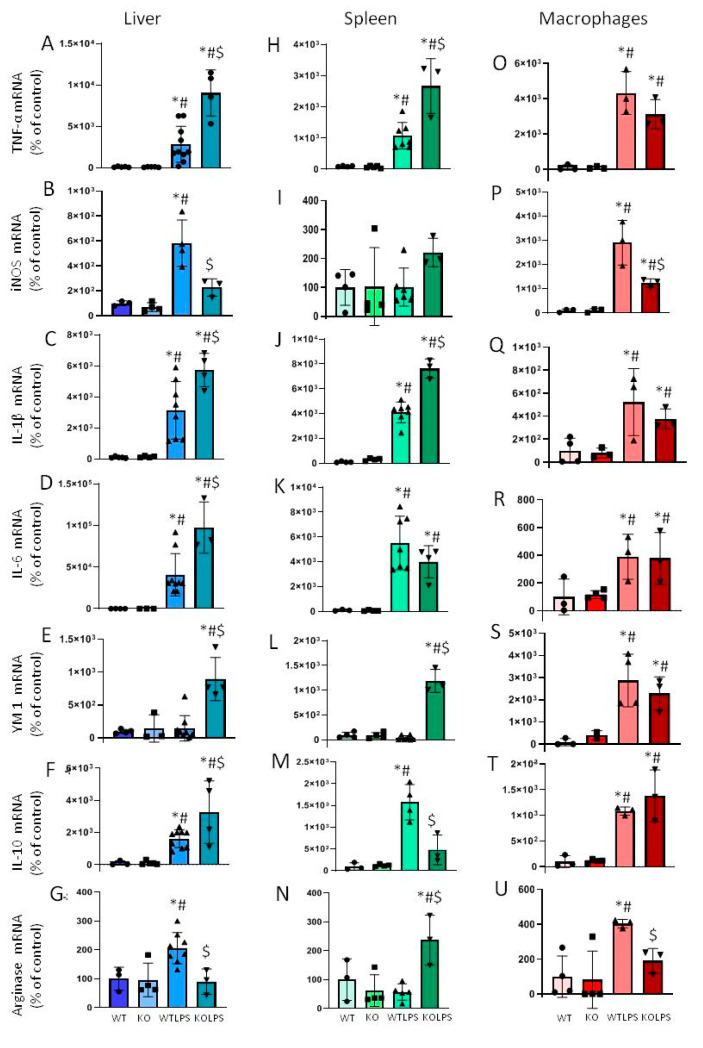
Effect of *Gal3* deletion on the expression of *TNF-α*, *iNOS*, *IL-1β*, *IL-6*, *YM1*, *IL-10*, and *arginase* mRNAs in the liver, spleen, and macrophages of mice from the different treatments assayed, measured by RT-PCR (**A**–**U**). Animals were culled 1 h after LPS/saline injection. Results are mean ± SD of *N* = 3–10 animals, normalized to *β-actin* and expressed as relative expression to the WT group. Statistical significance (one-way ANOVA followed by the LSD post hoc test for multiple comparisons): *—compared with WT group; #—compared with KO group; $—compared with WTLPS group; *p* < 0.001 for (**A**–**F**,**H**,**J**–**M**,**O**,**P**,**T**); *p* < 0.01 for (**G**,**N**,**S**); *p* < 0.05 for (**Q**,**R**,**U**). Abbreviations: WT, wild type mice; KO, Gal3 knockout mice; WTLPS, wild type mice treated with LPS; KOLPS, Gal3 knockout mice treated with LPS.

**Figure 7 ijms-23-01170-f007:**
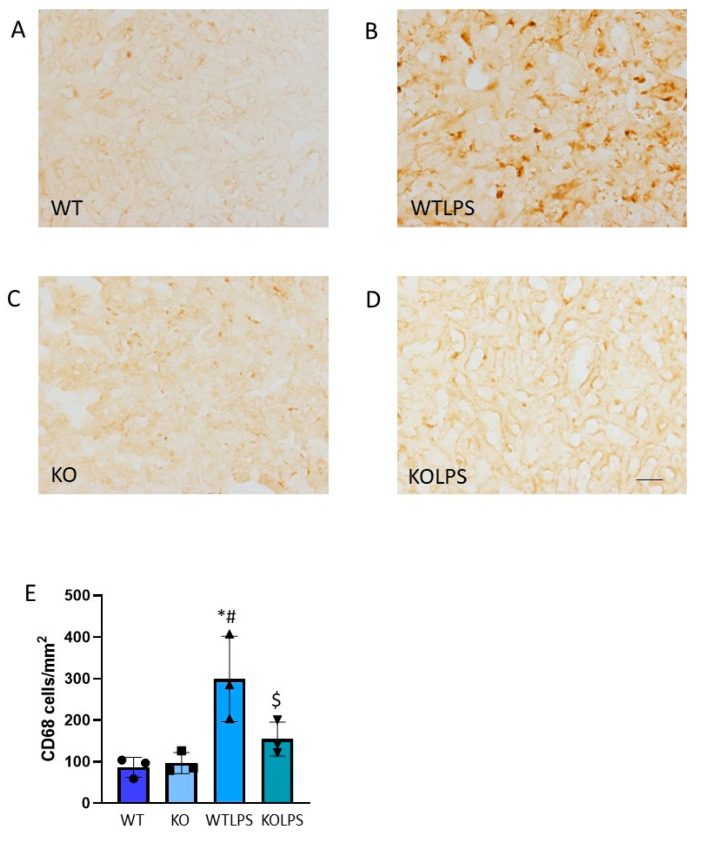
CD68 expression in liver. Representative immunostaining from sections of the different treatments assayed. WT (**A**) and KO (**C**) animals show a normal pattern of CD68 expression; however, the treatment with LPS produces a strong induction of CD68-positive cells in WT animals (**B**). Absence of *Gal3* clearly reduces this effect (**D**). Scale bar: 100 μm. (**E**) Quantification of the density of CD68 positive cells in liver. Results are mean ± SD of *N* = 3 animals, expressed as number of cells per mm^2^. Statistical significance (one-way ANOVA followed by the LSD post hoc test for multiple comparisons): *—compared with WT group; #—compared with KO group; $—compared with WTLPS group; *p* < 0.01. Abbreviations: WT, wild type mice; WTLPS, wild type mice treated with LPS; KO, Gal3 knockout mice; KOLPS, Gal3 knockout mice treated with LPS.

**Figure 8 ijms-23-01170-f008:**
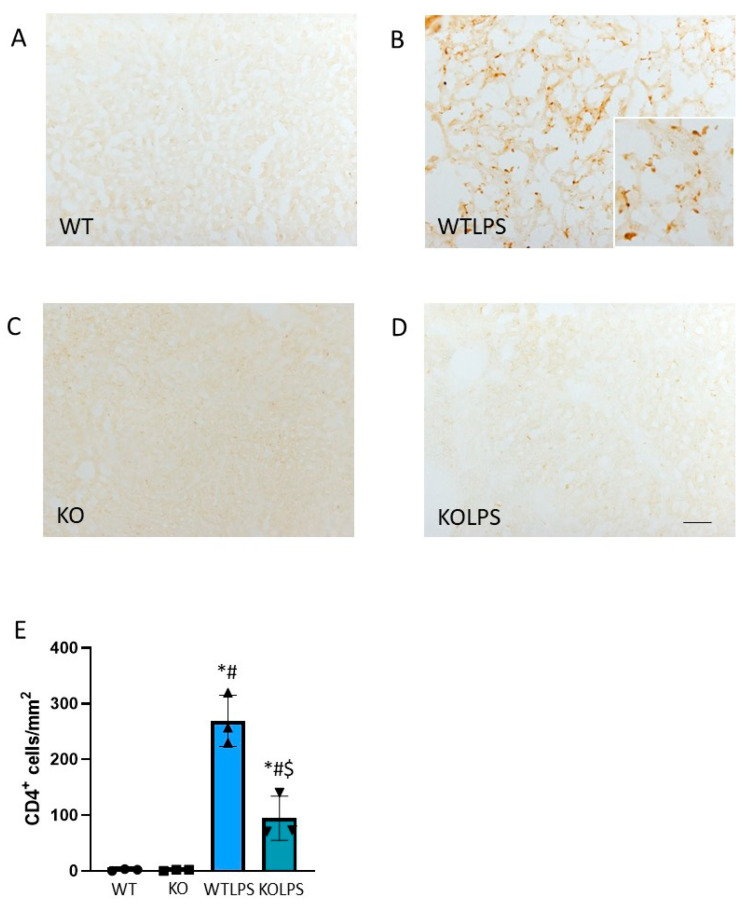
CD4 expression in liver. Representative immunostaining from sections of the different treatments assayed. WT (**A**) and KO (**C**) animals show a normal pattern of CD4 expression; however, the treatment with LPS produces a strong induction of CD4-positive cells in WT animals (**B**). Absence of *Gal3* clearly reduces this effect (**D**). Scale bar: 100 μm. (**E**) Quantification of the density of CD4 positive cells in liver. Results are mean ± SD of *N* = 3 animals, expressed as number of cells per mm^2^. Statistical significance (one-way ANOVA followed by the LSD post hoc test for multiple comparisons): *—compared with WT group; #—compared with KO group; $—compared with WTLPS group; *p* < 0.001. Abbreviations: WT, wild type mice; WTLPS, wild type mice treated with LPS; KO, Gal3 knockout mice; KOLPS, Gal3 knockout mice treated.

**Figure 9 ijms-23-01170-f009:**
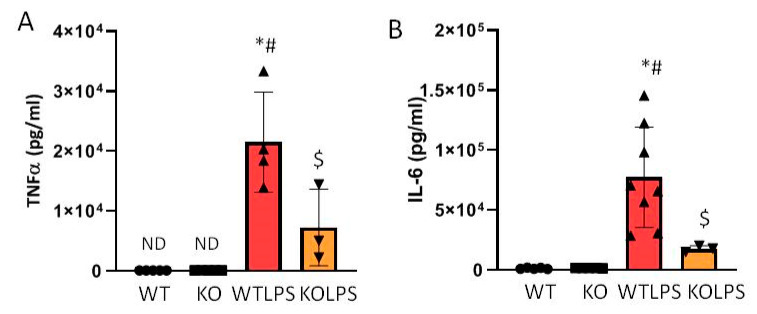
Determination of TNF-α and IL-6 levels in serum. TNF-α (**A**) and IL-6 (**B**) levels in serum of WT and Gal3KO animals. Blood samples were collected from the heart of mice 1 h after LPS/saline injection. Results are mean ± SD of *N* = 4–10 animals, expressed as ng/mLof the analyzed protein and relative to the WT group. Statistical significance (one-way ANOVA followed by the LSD post hoc test for multiple comparisons): *—compared with WT group; #—compared with KO group; $—compared with WTLPS group; *p* < 0.001. Abbreviations: WT, wild type mice; KO, Gal3 knockout mice; WTLPS, wild type mice treated with LPS; KOLPS, Gal3 knockout mice treated with LPS.

**Figure 10 ijms-23-01170-f010:**
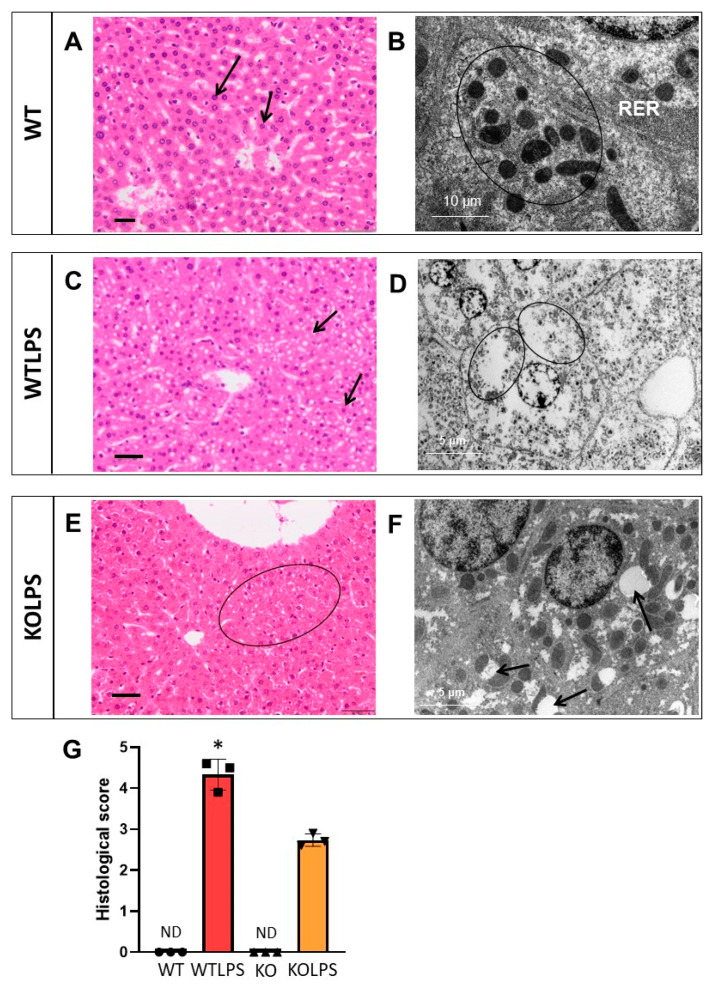
Histopathology of the livers of mice from the different treatments assayed. (**A**) Hepatic lobule with Remak cords of apparently normal hepatocytes (arrow). (**B**) Detail of hepatocyte with abundant rough endoplasmic reticulum (RER) and mitochondria (circle). (**C**) Hepatic parenchyma, appreciating abundant unilocular and multilocular steatosis (arrow). (**D**) Detail of hepatocyte with abundant diffuse fat (circle). (**E**) Detail of Remak cords with steatosis in the hepatocytes (circle). (**F**) Hepatocytes with vacuolations of its membranous system (arrow). (**G**) Histological score showing a semiquantitative analysis of steatosis in the liver. The pathology scores were as follows: 0, without significant injuries (0%); 1, minimum (<10%); 2, mild (11–25%); 3, moderate (26–50%); 4, marked (51–75%); 5, severe (>75%). Histopathological evaluation is performed in a blind manner by two highly experienced pathologists. Results are mean ± SD of *N* = 3 animals.Statistical significance (two tailed Student-*t* test): * *p* < 0.001. Abbreviations: WT, wild type mice; WTLPS, wild type mice treated with LPS; KO, Gal3 knockout mice; KOLPS, Gal3 knockout mice treated with LPS; ND, not damaged. (**A**,**C**,**E**), optical microscopy. Scale bars: 100 µm. (**B**,**D**,**F**), ultrastructural observations. Scale bars: (**B**), 10 µm; (**D**,**F**), 5 µm.

**Figure 11 ijms-23-01170-f011:**
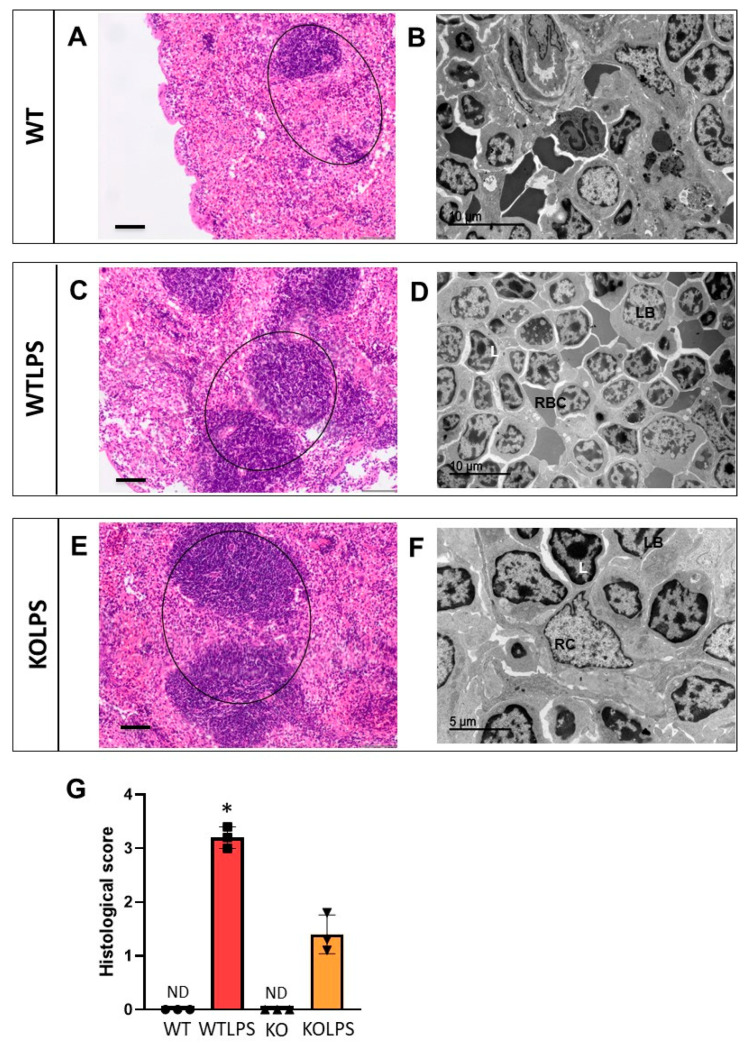
Histopathology of the spleens of mice from the different treatments assayed. (**A**) Apparently normal spleen. Normal white pulp (lymphoid follicles) (circle). (**B**) Marginal zone of the apparently normal spleen. (**C**) Hypertrophy and hyperplasia of lymphoid follicles very marked (circle). (**D**) Detail of lymphoid follicle with abundant lymphoblasts (LB) and lymphocytes (L), and some red blood cells (RBC). (**E**) Detail of spleen with hypertrophy and especially hyperplasia of the lymphoid follicles (circle). (**F**) Detail of lymphoid follicle with lymphocytes (L), lymphoblasts (LB) and reticular cells (RC). (**G**) Histological score showing a semiquantitative analysis of hypertrophy and hyperplasia of lymphoid follicles in the spleen. The pathology scores were as follows: 0, without significant injuries (0%); 1, minimum (<10%); 2, mild (11–25%); 3, moderate (26–50%); 4, marked (51–75%); 5, severe (>75%). Histopathological evaluation is performed in a blind manner by two highly experienced pathologists. Results are mean ± SD of *N* = 3 animals. Statistical significance (two tailed Student-*t* test), * *p* < 0.001. Abbreviations: WT, wild type mice; WTLPS, wild type mice treated with LPS; KO, Gal3 knockout mice; KOLPS, Gal3 knockout mice treated with LPS; ND, not damaged. (**A**,**C**,**E**), optical microscopy. Scale bars: 100 µm. (**B**,**D**,**F**), ultrastructural observations. Scale bars: (**B**,**D**), 10 µm; (**F**), 5 µm.

**Figure 12 ijms-23-01170-f012:**
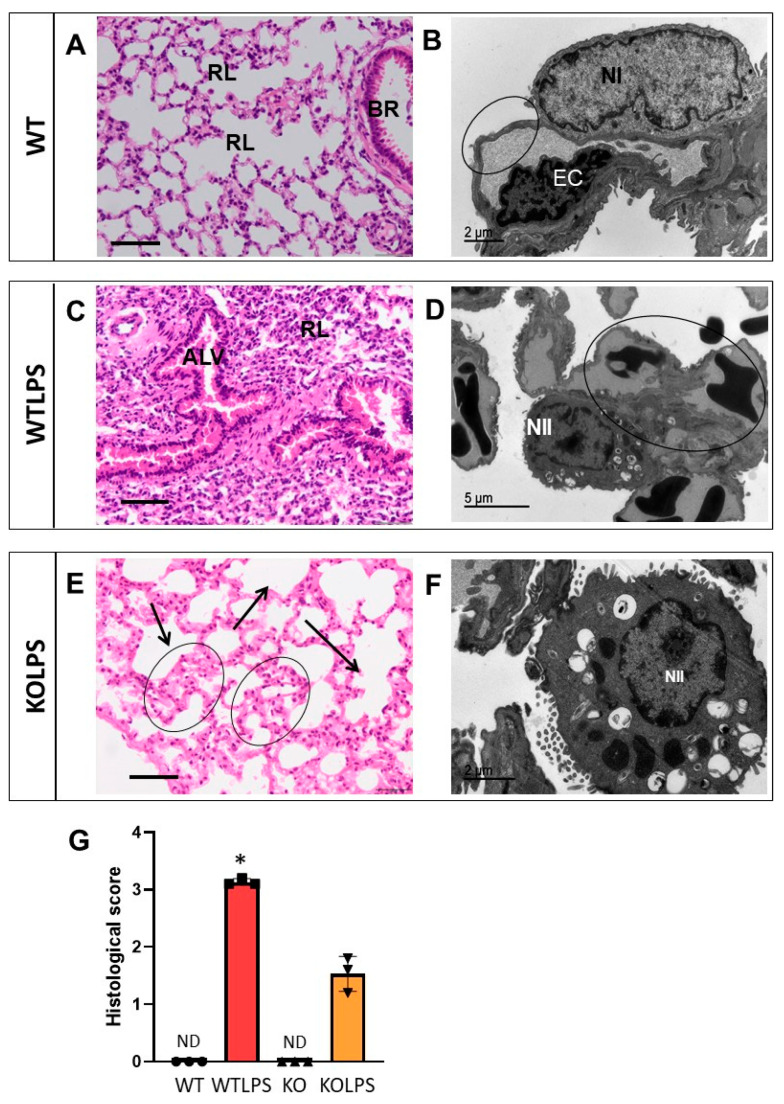
Histopathology of the lungs of mice from the different treatments assayed. (**A**) Lung in which an apparently normal bronchus (BR) and respiratory lobules (RL) stand out. (**B**) Detail of capillary and pulmonary alveolus separated by the respiratory barrier (circle) formed by the endothelial cell (EC) and pneumocyteI (NI). (**C**) Detail of lung that shows a marked atelectasis in both respiratory lobule (RL) and alveoli (ALV). (**D**) Detail of lung with capillary hyperemia (circle) and hypertrophy of pneumocytes II (NII). (**E**) Lung detail showing atelectasis zones (circle) and emphysema (arrow). (**F**) Septal hypertrophy of pneumocytes II (NII). (**G**) Histological score showing a semiquantitative analysis of atelectasis in the lung. The pathology scores were as follows: 0, without significant injuries (0%); 1, minimum (<10%); 2, mild (11–25%); 3, moderate (26–50%); 4, marked (51–75%); 5, severe (>75%). Histopathological evaluation is performed in a blind manner by two highly experienced pathologists. Results are mean ± SD of *N* = 3 animals. Statistical significance (two tailed Student-*t* test), * *p* < 0.001. Abbreviations: WT, wild type mice; WTLPS, wild type mice treated with LPS; KO, Gal3 knockout mice; KOLPS, Gal3 knockout mice treated with LPS; ND, not damaged. (**A**,**C**,**E**), optical microscopy. Scale bars: 100 µm. (**B**,**D**,**F**), ultrastructural observations. Scale bars: (**B**,**F**), 2 µm; (**D**), 5 µm.

**Figure 13 ijms-23-01170-f013:**
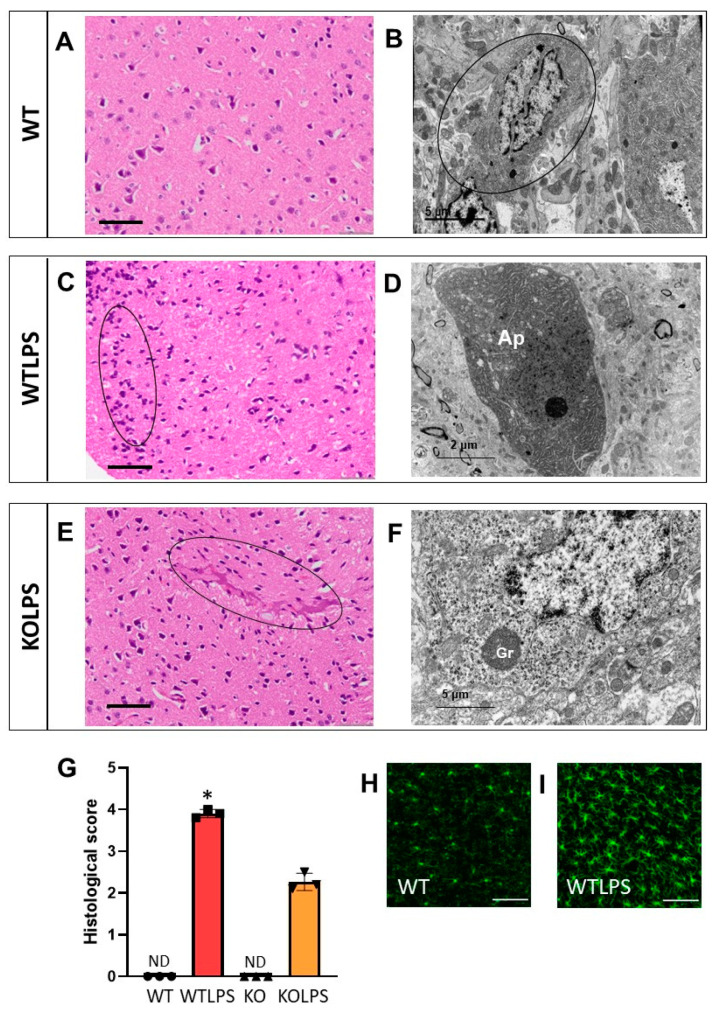
Histopathology of the brains of mice from the different treatments assayed. (**A**) Detail of cerebral cortex with abundant neurons. (**B**) Detail of apparently normal neuron and oligodendrocyte (circle). (**C**) Detail of cerebral cortex, showing mobilization of glia cells (circle). (**D**) Detail of a damaged neuron densified and vacuolized. (**E**) Detail of cerebral cortex with a tenuous hyperemia (circle). (**F**) Detail of neuron with lipofucsin precursor granules (Gr). (**G**) Histological score showing brain damage. The pathology scores were as follows: 0, without significant injuries (0%); 1, minimum (<10%); 2, mild (11–25%); 3, moderate (26–50%); 4, marked (5–75%); 5, severe (>75%). Histopathological evaluation is performed in a blind manner by two highly experienced pathologists. Results are mean ± SD of *N* = 3 animals. Statistical significance (two tailed Student-*t* test), * *p* < 0.001. Abbreviations: WT, wild type mice; WTLPS, wild type mice treated with LPS; KOLPS, Gal3 knockout mice treated with LPS; ND, not damaged. (**A**,**C**,**E**), optical microscopy. Scale bars: 100 µm. (**B**,**D**,**F**), ultrastructural observations. Scale bars: (**B**,**F**), 5 µm; (**D**), 2 µm. (**H**) Immunofluorescence of Iba1 showing a normal patter in microglial cells staining. (**I**) When animals were treated with LPS microglial cells activate and proliferate. Scale bars (**H**,**I**): 100 µm.

**Figure 14 ijms-23-01170-f014:**
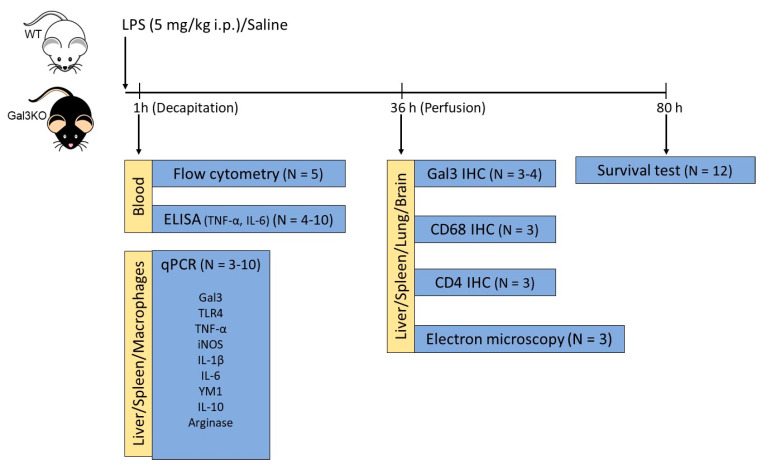
Timeline of the treatments with LPS in animals, and points at which the different parameters have been measured. WT and Gal3KO animals were injected with either LPS or saline and sacrificed at different time points depending on the technique assayed. The number of animals used in each experiment isalso indicated.

**Table 1 ijms-23-01170-t001:** Criteria for classifying endotoxemia according to the severity of the symptoms.

Score	Symptomatology
0	No symptoms
1	Reducedmobility and tremor
2	Reduced mobility, tremor, and prostration
3	Death

**Table 2 ijms-23-01170-t002:** Sense and antisense sequences of the primers used for the analysis of mRNA expression by qPCR.

Gene	Forward (5′–3′)	Reverse (5′–3′)
*β-actin*	5′-CCACACCCGCCACCAGTTCG-3′	5′-CCCATTCCCACCATCACACC-3′
*Gal3*	5′-GATCACAATCATGGGCACAG-3′	5′-GTGGAAGGCAACATCATTCC-3′
*TNF-α*	5′-TGCCTATGTCTCAGCCTCTTC-3′	5′-GAGGCCATTTGGGAACTTCT-3′
*iNOS*	5′-CTTTGCCACGGACGAGAC-3′	5′-TCATTGTACTCTGAGGGCTGAC-3′
*Arginase*	5′-TCACCTGAGCTTTGATGTCG-3′	5′-CTGAAAGGAGCCCTGTCTTG-3′
*YM1*	5′-GTACCCTGGGTCTCGAGGAA-3′	5′-GCCTTGGAATGTCTTTCTCAC-3′
*IL-6*	5′-GACAAAGCCAGAGTCCTTCAGA-3′	5′-AGGAGAGCAATTGGAAATTGGGG-3′
*IL-1β*	5′-TGTAATGAAAGACGGCACACC-3′	5′-TCTTCTTTGGGTATTGCTTGG-3′
*IL 10*	5′-CCAAGCCTTATCGGAAATGA-3′	5′-TTTCACAGGGAGAAATCG-3′
*TLR-4*	5′-GCCTCGAATCCTGAGCAAAC-3′	5′-CTCTCGGTCCATAGCAGAGC

Abbreviations: *Gal3*, galectin-3; *TNF-α*, tumor necrosis factor α, *iNOS*, inducible nitric oxide synthase; *YM1*, protein type chitinase 3; *IL*, interleukin; *TLR4*, Toll-like receptor 4.

## Data Availability

The datasets generated during and/or analyzed during the current study are available from the corresponding author on reasonable request.

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
