# Peer review of "Gal3 Plays a Deleterious Role in a Mouse Model of Endotoxemia"

_ijms, 2022, doi:10.3390/ijms23031170_

Round 1

Reviewer 1 Report

Manuscript by Juan Carlos Fernandez-Martín ‘Gal3 plays a deleterious role in a mouse model of endotoxemia’ dealing with the role of Gal3 in endotoxemia induced immune response. Study results are clear, and discussion is also seeming good but in results and discussion section too many things are over discussed instead of concise writing. Authors was not able to corelate results in discussion section clearly.

Authors should discuss the adverse effect of Gal3 KO whether this mechanism of modulating immune response is indirect or direct at molecular level? In introduction section.

For these type of studies it would be better to make a chart of treatment plan with number and type of animals like WT KO ….control etc. like when they sacrificed animals and how.

Fig.2 related Line 132 -136…statement is confusing please write clear statement.

What are the normal levels of Gla3 in human peripheral blood and in mice? Are there other conditions except immune response where Gal3 play important role?

Why authors did not estimate Gal3 levels in serum after 36 hr of LPS treatment?? And why estimated at 1 hr?

In Fig4. Authors should put either histo images or bar graph of liver, spleen and lungs. Better to put images along with quantification rest on authors.

Fig5. Authors should correct legends mRNA fold change or delta delta ct etc………. Please check in all graphs.

Do authors have any invitro data regarding the use of Gal3 inhibitors in invitro immune cell model system to proof this hypothesis?

Authors should make all bar graphs with dot plots.

Why authors choose only male animals over female in this study while they used both in assessing LPS dosing vs survival experiments. What are the levels of Gal3 in female vs male in survival experiments? in all the set.

Author Response

Responses to the reviewers

Reviewer 1

Manuscript by Juan Carlos Fernandez-Martín ‘Gal3 plays a deleterious role in a mouse model of endotoxemia’ dealing with the role of Gal3 in endotoxemia induced immune response. Study results are clear, and discussion is also seeming good but in results and discussion section too many things are over discussed instead of concise writing. Authors was not able to corelate results in discussion section clearly.

Answer: We sincerely thank this reviewer for his/her overall positive valuation of our work and careful revision of our manuscript and constructive comments.

  1. Authors should discuss the adverse effect of Gal3 KO whether this mechanism of modulating immune response is indirect or direct at molecular level? In introduction section.

Answer: To clarify whether the modulation of immune cells by Gal3 is a direct effect at the molecular level, the expression of TRL4, a Gal3 ligand, was measured in macrophages, liver, and spleen 1 hour after LPS injection in WT and GAL3KO mice. This analysis demonstrates a nice effect of Gal3 deletion on TLR4 expression in macrophages as it prevents the LPS-induced expression of this receptor (Figure 3).  It is known that TLR4 is one of the main drivers of macrophage activation, triggering several transduction pathways, such as the nuclear factor kappa B (NF-κB) pathway and mitogen-activated protein kinases (MAPKs) pathway, which cause increased expression of inflammatory cytokines. Therefore, our results show that the effects on immune responses driven by Gal3 could be mediated by TLR4 and its downstream signaling cascade. This information is now clearly stated in the discussion section (Line 494-504)

  1. For these type of studies it would be better to make a chart of treatment plan with number and type of animals like WT KO ….control etc. like when they sacrificed animals and how.

Answer: We have followed this reviewer´s suggestion and a new figure (New Figure 14) explaining the different treatments is included in the new version of the manuscript.

  1. Fig.2 related Line 132 -136…statement is confusing please write clear statement.

Answer: Following this reviewer's suggestion, we have changed the sentence to make it clearer (Lines 141-145).

  1. What are the normal levels of Gla3 in human peripheral blood and in mice? Are there other conditions except immune response where Gal3 play important role?

Answer: Normal Gal3 serum levels in humans are around 8 ng/ml. Normal Gal3 serum levels in mice are shown in Fig. 3 (about 250.000 pg/ml). Regarding the second question, there are several conditions in which Gal3 plays important roles. For instance, Gal3 has many physiological functions, including cell-to-cell interactions, adhesion, apoptosis, and cell cycle regulation. Moreover, the role of Gal3 in several diseases such as cancer and cardiac, hepatic, fibrotic and periodontal diseases have been described. This information is now included in the Introduction section (Line 62-68).

  1. Why authors did not estimate Gal3 levels in serum after 36 hr of LPS treatment?? And why estimated at 1 hr?

Answer: We have previously reviewed the literature on the best time point for our experiment and found that most studies with serum samples are performed in the first 1-2 hours after LPS treatment (i.e. PMID: 17203472). This is due to the rapid change in serum levels of cytokines and other inflammatory mediators that occurs after LPS challenge. Therefore, we decided to collect blood samples during the first hour, and consequently, all measurements made in blood samples follow this time point. In fact, at this point we have found changes in Gal3 levels in serum, which made us think that we have chosen a good time point. Importantly, 1 hour is not enough time to produce tissue damage. Therefore, we have divided the experiment into two different time points: 1 hour for blood samples and 36 hours for tissue samples.

  1. In Fig4. Authors should put either histo images or bar graph of liver, spleen and lungs. Better to put images along with quantification rest on authors.

Answer: We are not sure if we have understood this reviewer's comment. We think that the referee suggests including all histological images from liver, spleen and lung. Therefore, we have designed a new figure 4 including all photographs and bar graphs.

  1. Fig5. Authors should correct legends mRNA fold change or delta delta ct etc………. Please check in all graphs.

Answer: We agree with this reviewer´s comment and the legend has been modified (now figure 6). As stated in the figure legend, results are mean ± SD of N animals normalized to β-actin and expressed as relative expression to the WT group. Therefore, we have included “% of control” after the name of the cytokine in the legend of Y axis. Example: iNOS mRNA (% of control).

  1. Do authors have any in vitro data regarding the use of Gal3 inhibitors in in vitro immune cell model system to proof this hypothesis?

Answer: In previously published papers our group have used two experimental approaches to inhibit Gal3 in vitro (BV2 microglial cell line): Gal3 expression was suppressed using siRNA and Gal3 blocking antibody was used to neutralize the effects of released Gal3. In this study we observed that both methods highly prevented the LPS-induced inflammation. Moreover, to validate Gal3 effect over the inflammatory response upon LPS stimulus, the release of several proinflammatory cytokines were checked in WT and Gal3KO primary cell cultures, confirming the BV2 cell data, with reduced inflammatory response in Gal3KO microglia. This information is included in the discussion section (Line 536-539).

Ref: Microglia-Secreted Galectin-3 Acts as a Toll-like Receptor 4 Ligand and Contributes to Microglial Activation. Burguillos MA, Svensson M, Schulte T, Boza-Serrano A, Garcia-Quintanilla A, Kavanagh E, Santiago M, Viceconte N, Oliva-Martin MJ, Osman AM, Salomonsson E, Amar L, Persson A, Blomgren K, Achour A, Englund E, Leffler H, Venero JL, Joseph B, Deierborg T. Cell Rep. 2015, 10(9):1626-1638. doi: 10.1016/j.celrep.2015.02.012.

  1. Authors should make all bar graphs with dot plots.

Answer: We have followed this reviewer's suggestion and all bar graphs have been changed by bar graphs with dot plots.

  1. Why authors choose only male animals over female in this study while they used both in assessing LPS dosing vs survival experiments. What are the levels of Gal3 in female vs male in survival experiments? in all the set.

Answer: We did not want to mix female and male in this study since the anti-inflammatory effects of estrogens are well known and this new variable could be the origin of a higher deviation and lower reproducibility. Survival curve in female is only very preliminary data made to tune up the best dose of i.p. LPS. Once we had an idea of the effects that the i.p. injection of LPS made on mice, we performed the same dose-response curve in male, and then the rest of the experiments were made only in males. We have included all our preliminary dose-response curves, but in order to avoid the possible confusion that this data could have in the reading of the paper, female data are not included in this version of the manuscript.

Reviewer 2 Report

In this manuscript, Fernandez-Martin et al. demonstrated the role of Gal-3 in murine model of endotoxemia. The authors demonstrated that genetic ablation of Gal-3 protects against LPS induced endotoxemia and surface expressing/binding of Gal-3 was inhibited on the peripheral organs like liver and spleen. Further they have demonstrated that Gal-3 KO mice exhibit ameliorated clinical symptoms and reduced tissue damage upon LPS administration as compared to WT controls.

However, I have the following comments.

  1. This study showed that Gal-3 KO mice exhibit protection against LPS induced endotoxemia in a subclinical dose whereas a high dose (7.5 mg/Kg) abolished the effect in agreement with a previous study by Li et.al. The authors should explain the mechanism behind this phenomenon.
  2. Does the Gal-3 expression on the surface of different immune cells down regulated after in-vitro stimulation of LPS?
  3. The authors demonstrated that expression of Gal-3 (Fig-4) on liver and lung increased upon LPS administration. Does the same phenomenon happen in liver, lung and spleen immune cells?
  4. The authors demonstrated that there is an increased inflammatory cytokine such as TNF-α, IL-1β, IL-6 mRNA in Gal-3 KO mice whereas the protein levels of TNF-α, IL-1β, IL-6 significantly less. It has been believed that early inflammatory response is detrimental for sepsis. In contrast the authors claimed that Gal-3 KO mice were protected against LPS induced endotoxemia. This need to be explained.
  5. The authors should administer anti-Gal-3 antibody and show its effect on LPS induced endotoxemia.

Minor comments

  1. The labeling of figure numbers such as A, B etc. should be outside of the panel.
  2. The letters such as a, ab, c in the figure were not mentioned in the text.
  3. The authors should show untreated KO group in fig. 9-12.

Author Response

Reviewer 2

In this manuscript, Fernandez-Martin et al. demonstrated the role of Gal-3 in murine model of endotoxemia. The authors demonstrated that genetic ablation of Gal-3 protects against LPS induced endotoxemia and surface expressing/binding of Gal-3 was inhibited on the peripheral organs like liver and spleen. Further they have demonstrated that Gal-3 KO mice exhibit ameliorated clinical symptoms and reduced tissue damage upon LPS administration as compared to WT controls.

Answer: We also sincerely thank this reviewer for his/her careful reading of our manuscript and constructive comments.

However, I have the following comments.

  1. This study showed that Gal-3 KO mice exhibit protection against LPS induced endotoxemia in a subclinical dose whereas a high dose (7.5 mg/Kg) abolished the effect in agreement with a previous study by Li et.al. The authors should explain the mechanism behind this phenomenon.

Answer: The article mentioned by this reviewer (Li et al., J Immunol 2018; PMID: 18684969) was extensively discussed in our previous version. In this study, the authors claimed that Gal3 acts as a negative regulator of LPS-induced endotoxemia and inflammation, in sharp contrast to our study. According to this paper, animals lacking Gal3 succumbed earlier to systemic LPS and exhibited higher serum TNF-α levels after LPS challenge. In the same study, Gal3 deletion exacerbated LPS-induced inflammatory response of bone marrow derived macrophages. The authors found that Gal3 binds to LPS, thus inhibiting the LPS-associated pro-inflammatory response as tested in bone marrow-derived macrophages (please, see discussion section, lines 505-553). However, the dose used by Li et al. was not 7.5 mg/Kg but 23 mg/Kg.  A plausible explanation for the quite contrasting effects of Gal3 on LPS-associated endo-toxemia found by Li et al. (2008) and our study may rely on the LPS doses used in each study. The lethal dose 50 (LD50) of LPS is about 1-25 mg/kg (Fink, 2014). While Li et al. used a LPS dose of 23 mg/kg (upper limit), we used a dose of 5 mg/kg (lower limit). These great differences in LPS doses may certainly be behind the quite opposite effects (beneficial vs detrimental) ascribed to Gal3 under conditions of endotoxemia. We would like to state that we have long used both LPS and gal3 in immune cells and we always found that Gal3 synergistically amplifies LPS-induced immune responses. These studies have been published in excellent journals including Acta Neuropathologica (PMID: 31006066), Cell Reports (PMID: 25753426), Scientific Reports (PMID: 28128358) and Acta Neuropathologica Communications (PMID: 25387690). In our opinion, these studies (and others in the literature; see discussion section) exclude that Gal3 may inhibit LPS action by binding to it.

  1. Does the Gal-3 expression on the surface of different immune cells down regulated after in-vitro stimulation of LPS?

Answer: As far as we know there is a study showing the effects of LPS on THP-1 cells, a monocyte/macrophage cell line. In these cells, the amount of Gal3 on the cell surface is not detectable before and after LPS treatment.

3.The authors demonstrated that expression of Gal-3 (Fig-4) on liver and lung increased upon LPS administration. Does the same phenomenon happen in liver, lung and spleen immune cells?

Answer: To test this hypothesis we have made a double immunofluorescence with Iba1 (a monocyte/macrophage marker) and galectin-3 and found that most Gal3+ cells are Iba1+. Therefore, we can conclude that expression of Gal3 increases in immune cells upon LPS administration. This information is now included in new Figure 5. Material and Methods, Results and Discussion sections have been modified accordingly.

  1. The authors demonstrated that there is an increased inflammatory cytokine such as TNF-α, IL-1β, IL-6 mRNA in Gal-3 KO mice whereas the protein levels of TNF-α, IL-1β, IL-6 significantly less. It has been believed that early inflammatory response is detrimental for sepsis. In contrast the authors claimed that Gal-3 KO mice were protected against LPS induced endotoxemia. This need to be explained.

Answer: Regarding the PCR analysis, this is a clear example sustaining the complexity of tissue-specific immune responses and we clearly stated in the discussion section that this analysis was not enough for drawing any conclusions about the pro-inflammatory role of Gal3 in our model of endotoxemia in the different organs analyzed. A quite plausible explanation that could explain this apparent discrepancy is related to the fact that time-associated cytokine changes are very fast and transient. This is well illustrated in a previous study that performed a time course of cytokine levels in liver, plasma and brain following 5 mg/kg LPS (PMID: 17203472). This analysis demonstrated that in a very narrow time window (2 hrs), liver TNF reached a peak in 30 min-1 hr to drop 30 min later. Indeed, this study suggests:  i) the time point chosen in our study (1 hr) is quite reasonable and ii) cytokine levels are very transient and consequently, their levels in each experimental condition has to be considered cautiously. This information is now included in the Discussion section (Lines 521-523).

  1. The authors should administer anti-Gal-3 antibody and show its effect on LPS induced endotoxemia.

We agree with the referee that it would be very interesting to probe the effect of Gal3 inhibitors in our experimental conditions. However, we have to keep in mind that these experiments would require testing different doses of the commercial inhibitors and study this effect on our particular experimental conditions, which would take a long time and use many new animals.

Minor comments

  1. The labeling of figure numbers such as A, B etc. should be outside of the panel.

Answer: We have changed the labeling of figure numbers as suggested.

  1. The letters such as a, ab, c in the figure were not mentioned in the text.

Answer: These letters refer to the statistical analysis, as stated in the figure legend. We honestly think that including the letters also in the results section could make the interpretation of the data more difficult, since after each result we have included the number and letter of the figure and the statistical significance. However, if the reviewer considers this point is absolutely necessary, we would include it in this section as well.

  1. The authors should show untreated KO group in fig. 9-12.

Answer: Photographs showing KO animals are included in Supplementary Figure 8. We decided this distribution since KO animals injected with saline have virtually the same features that WT animals injected with saline. Including these pictures in figures 9-12 could make us reduce the size of the rest of the pictures making it difficult to appreciate the details we want to show in these figures. However, if this referee considers this point essential, we would be happy to change it

Round 2

Reviewer 1 Report

Authors addressed most of the comments while manuscript still have few general issues.

1. I did not find answer of comment 8 in revised manuscript. While its showing in author response.

comment 8. Do authors have any in vitro data regarding the use of Gal3 inhibitors in in vitro immune cell model system to proof this hypothesis?

Answer: In previously published papers our group have used two experimental approaches to inhibit Gal3 in vitro (BV2 microglial cell line): Gal3 expression was suppressed using siRNA and Gal3 blocking antibody was used to neutralize the effects of released Gal3. In this study we observed that both methods highly prevented the LPS-induced inflammation. Moreover, to validate Gal3 effect over the inflammatory response upon LPS stimulus, the release of several proinflammatory cytokines were checked in WT and Gal3KO primary cell cultures, confirming the BV2 cell data, with reduced inflammatory response in Gal3KO microglia. This information is included in the discussion section (Line 536-539).

2. Authors should improved Fig 5 quality.Fig 5 is not looking standard for publication.

This time authors can send response directly to editor.

Accept after revision. 

Author Response

Authors addressed most of the comments while manuscript still have few general issues.

Answer: We thank again this referee for his/her careful reading of our manuscript.

  1. I did not find answer of comment 8 in revised manuscript. While its showing in author response.

comment 8. Do authors have any in vitro data regarding the use of Gal3 inhibitors in in vitro immune cell model system to proof this hypothesis?

Answer: In previously published papers our group have used two experimental approaches to inhibit Gal3 in vitro (BV2 microglial cell line): Gal3 expression was suppressed using siRNA and Gal3 blocking antibody was used to neutralize the effects of released Gal3. In this study we observed that both methods highly prevented the LPS-induced inflammation. Moreover, to validate Gal3 effect over the inflammatory response upon LPS stimulus, the release of several proinflammatory cytokines were checked in WT and Gal3KO primary cell cultures, confirming the BV2 cell data, with reduced inflammatory response in Gal3KO microglia. This information is included in the discussion section (Line 536-539).

Answer: Actually, this information was included in the revised manuscript, but it was not marked in red. We have now included all our response and marked it in red (lines 536-543)

  1. Authors should improved Fig 5 quality.Fig 5 is not looking standard for publication.

Answer: We have changed this figure by new pictures. Background has clearly decreased and this has improved the quality of the pictures